# Myocardium miRNA Analysis Reveals Potential Biomarkers of Sudden Coronary Death in Rats

**DOI:** 10.3390/cimb47110889

**Published:** 2025-10-28

**Authors:** Chunmei Zhao, Xinyu Zhou, Yaqin Bai, Zhenxiang Zhao, Huaping Zhang, Cairong Gao, Keming Yun, Xiangjie Guo

**Affiliations:** 1School of Forensic Medicine, Shanxi Medical University, Jinzhong 030606, China; zcmm1001@163.com (C.Z.); 15392959696@163.com (X.Z.); byqlzp@163.com (Y.B.); gaocairong@sxmu.edu.cn (C.G.); 2Translational Medicine Center, Shanxi Medical University, Jinzhong 030606, China; zhaozhenxiang@sxmu.edu.cn (Z.Z.); zhanghuaping@sxmu.edu.cn (H.Z.); 3Key Laboratory of Forensic Toxicology, Ministry of Public Security, Jinzhong 030606, China

**Keywords:** forensic pathology, cause of death, sudden coronary death, miRNA expression, rats, next-generation sequencing

## Abstract

This study aims to provide potential biomarkers and reveal the molecular mechanism of sudden coronary death (SCD). Rat models of atherosclerotic death (ASD), coronary atherosclerosis (AS), and acute myocardial ischemia (AMI) and sham groups were established via the gavage of high-fat emulsion and left coronary artery ligation. The myocardium was collected, and transcriptome sequencing was performed. Differentially expressed miRNAs (DEmiRNAs) were identified using edeR software. The target genes were predicted using TargetScan, and functional enrichment analysis was performed via KEGG. Then, an miRNA–mRNA interaction network was constructed using Cytoscape. The key miRNAs with biomarker potential were identified using LASSO regression. A total of 217, 224, and 86 DEmiRNAs were identified in the ASD, AS, and AMI groups compared with the sham group, respectively. The Ras and Rap1 pathways were mainly expressed in ASD. The β-alanine and sphingolipid metabolisms were expressed in AMI. Finally, miR-106b, miR-195, miR-33, miR-652, miR-466b, and miR-6321 were identified as biomarkers of ASD. MiR-205, miR-877, miR-325, and miR-344b were identified as biomarkers of AMI. miR542-Atg12 was involved in the RIG-I-like receptor signaling pathway, miR6328-Gstz1 was involved in tyrosine metabolism, and miR483-Dusp5 was involved in the MAPK signaling pathway. This study provides a reference for the identification of SCD in forensic pathology.

## 1. Introduction

Sudden cardiac death is defined as unexpected death due to an underlying cardiac disease [1,2] and is a major public health problem globally. Worldwide, the leading causes of sudden cardiac death are coronary atherosclerotic disease (CAD) and acute myocardial ischemia (AMI), which account for 70–80% of sudden cardiac death cases [3,4]; this condition is also known as sudden coronary death (SCD). CAD is characterized by the existence of atherosclerosis (either obstructive or non-obstructive) in the coronary arteries [5]. The main mechanism of CAD-SCD is atherosclerosis, which can be complicated by intraplaque hemorrhage, thrombosis, and severe coronary artery stenosis, which causes myocardial ischemia and eventually leads to arrhythmia and death. However, some cases without atherosclerosis or with mild lesions can also cause SCD, such as coronary artery spasm (CAS) and muscle bridge (MB). In the practice of forensic pathology, some autopsies present atheromatous lesions but not to the extent of death or show the coexistence of lesions and other diseases. Some autopsies are negative after the completion of the microscopic autopsy and other laboratory tests [6]. Therefore, accurately diagnosing SCD has become an important task of forensic pathology.

Ideally, biomarkers should meet the following criteria: sensitive, specific, and easily obtainable [7]. For many years, scholars have been searching for stable and sensitive methods and biomarkers for the diagnosis of SCD. This study showed that the values for picolinic acid/Kynurenic acid and picolinic acid/3-hydroxyanthranillic acid in the CHD group were higher than in the control group, thus providing a better choice of biomarker for SCD [8]. Allen et al. found that the hs-C-reactive protein is significantly elevated in SCD, enabling it to act as a biomarker of SCD [9]. However, due to cadaver autolysis and spoilage, these biomarkers still have certain limitations. We urgently need to find specific biomarkers related to SCD.

miRNAs are non-protein-coding RNA molecules that participate in post-transcriptional regulation, about 21–22 nt long [10,11]. MiRNAs have better stability compared to proteins and metabolites. MiRNA cannot easily be degraded by ribonuclease and are resistant to changes in environmental factors such as temperature and pH [12]. MiRNAs regulate gene expression by binding to complementary sequences on target mRNAs, primarily within the 3 ′-untranslated region (3 ′UTR) or promoter regions [13]. This typically leads to either the degradation of the target mRNA or the inhibition of its translation into protein. The interaction of miRNAs and mRNAs enables miRNAs to effectively modulate biological processes and maintain homeostasis [14]. In forensic practice, miRNA can be used for the identification of tissues and body fluids, cause-of-death analysis, time-related estimation, age estimation, and the identification of monozygotic twins [15,16,17]. At present, research mainly focuses on the relationship between miRNA and CAD. Overexpression of miR-30c can reduce lipid synthesis and alleviate atherosclerotic lesions in ApoE-deficient mice [18]. Few studies on the differential expression of miRNA in SCD have been conducted.

To summarize, this study aims to establish an SCD rat model and analyze the DEmiRNAs in the myocardium. By identifying key miRNAs and constructing an miRNA-mRNA network in SCD, we present a promising predictive biomarker for the forensic identification of SCD.

## 2. Materials and Methods

### 2.1. Animals

SPF-grade male Sprague Dawley rats (200 ± 20 g, 4–6 weeks) were obtained from the Animal Center of Beijing. The rats were kept in a 12 h:12 h alternating light and dark environment, with relative humidity of 40–70% and an ambient temperature of 22 ± 2 °C. All rats were kept under specific pathogen-free conditions in the room and given free access to food and water. The rat experiment performed in this study was carried out after one week of acclimatization. The 24 rats were randomly and equally divided into four groups: the atherosclerotic death (ASD), coronary atherosclerosis (AS), acute myocardial ischemia (AMI), and sham group. The study was conducted in accordance with the Declaration of Helsinki, and all animal experiments were performed in accordance with international, national, and institutional guidelines for animal care.

### 2.2. Surgical Modeling

Rats in the AS group were subjected to the intraperitoneal injection of a high-fat emulsion combined with vitamin D3 for twelve weeks. Rats in the AMI group were subjected to the ligation of the left anterior descending coronary artery until 12 weeks. Rats in the ASD group were subjected to the high-fat emulsion and coronary artery ligation. The sham rats were subjected to threading without ligation until 12 weeks and then euthanized via an overdose of anesthesia. The specific method of modeling is as described earlier [19]. Blood samples were taken from the orbit to measure the concentrations of blood lipids using an AU5800 automatic biochemical analyzer (Beckman Coulter Company, Brea, CA, USA), and the atherosclerosis index was calculated to prove the presence or absence of atherosclerosis. Myocardial infarction was observed via an electrocardiogram. The left ventricular myocardium was extracted from each group and used for next-generation sequencing and HE staining.

### 2.3. Sample Extraction and Preparation

About 100 mg of left ventricular myocardium from each rat was extracted using sterile surgical instruments and immediately frozen in liquid nitrogen. It was then stored at −80 °C for RNA sequencing analysis. The remaining left ventricular myocardium in each group was fixed in 4% paraformaldehyde solution at room temperature and stored there for hematoxylin–eosin (HE) staining and histological observation.

### 2.4. RNA Isolation and Sequencing

The total RNA of the heart tissue of rats was extracted using TRIzol reagent (Thermo Fisher Scientific Company, Waltham, MA, USA.) according to the manufacturer’s protocol. An enzyme-labeled instrument (Infinite M200 Pro, Tecan company, Männedorf, Switzerland) was used to detect the purity (OD260/OD280 = 1.8–2.2) and concentration of the RNA samples. The samples with an RNA integrity number (RIN) value greater than 8.0 were selected. The library was constructed once the samples had been qualified. Then, the q-PCR method was used to accurately quantify the effective concentration of the library to ensure the quality of the library. The samples were then sequenced using the Nova Seq 6000 Sequencing System (Illumina Company, San Diego, CA, USA).

### 2.5. HE Staining

The myocardium tissues were fixed in 10% formalin, processed, and embedded in paraffin. 4 μM thick were dewaxed and rehydrated through an ethanol gradient. The slice was stained with hematoxylin and eosin. Then the slice was subjected to dehydration and clearing processes.

### 2.6. MiRNA Data Preprocessing

The original miRNA data was converted into an original sequencing sequence through base identification. The final data was stored in FASTQ format. The data was quality-controlled to ensure the accuracy of the sequencing information. Finally, a high-quality sequence was obtained for subsequent bioinformatics analysis:(1)The joint was removed.(2)Sequences shorter than 18 nucleotides or longer than 30 nucleotides were removed.(3)Reads with more than 10% unknown bases were removed.(4)Sequences with a low quality value for each sample were removed.

### 2.7. Identification of miRNAs

The sequences aligned to the reference genome were aligned with the mature sequences of known miRNAs in the miRBase (v22) database and their upstream and downstream ranges of 2 nt and 5 nt, with up to one mismatch allowed. The sequences thus identified were considered to be known miRNAs. For the sequences of unknown miRNAs, miRDeep2 software (v2.0.5) was used to predict new miRNAs.

### 2.8. Bioinformation Analysis

The expression of miRNA in each rat sample was normalized using the TPM algorithm. Then, a volcano plot and heat map were drawn for the differential miRNAs between the groups by edgeR software (v3.8.6), which used the following screening criteria: |log2(FC)| ≥ 0.58 and corrected *p* value ≤ 0.05. Target gene prediction was performed using miRanda (v3.3a) and TargetScan (v5.0) based on the gene sequence information. The BLAST software (v2.2.26) was used to obtain annotation information concerning the target genes. KEGG pathway enrichment analysis was performed on the target genes of differentially expressed miRNAs. LASSO regression was used to screen key miRNAs. Finally, an miRNA-mRNA regulatory network was constructed using Cytoscape (v3.8.2). The ROC curve was drawn for the miRNAs selected by Lasso regression.

### 2.9. Statistical Analysis

Data are shown as means ± SEMs. Statistical analysis was performed using GraphPad Prism 7 software. In experiments comparing two groups, statistical differences between groups were determined using a Student *t* test. Differences were considered significant at ∗ *p* < 0.05. The Benjamini–Hochberg correction method was used to adjust the original hypothesis test results. Finally, the False Discovery Rate (FDR) was used to screen for differentially expressed miRNAs.

## 3. Results

### 3.1. Successful Development of Rat SCD Model

To explore whether the atherosclerosis model was successfully established, we estimated four lipid parameters in the serum: total cholesterol (TC), Triglyceride (TG), high-density lipoprotein (HDL), and low-density lipoprotein (LDL). It showed that TC, TG, and LDL were significantly increased and that HDL was decreased in the AS and ASD group (Table 1). The atherosclerosis index (AI = (TC − HDL)/HDL) is a clinical measure used to quantify the degree of arterial stiffness. AI ≥ 4 indicated the presence of significant atherosclerotic lesions. The AI was 4.77 in the AS and ASD group, indicating that coronary atherosclerosis was successfully established. We found that endothelial cells were disordered, that the intima was thickened, that the lumen narrowed (Figure 1A,B), and that perivascular inflammatory cells infiltrated (Figure 1A,C). We also found that the myocardium showed necrosis of the contraction zone and the infiltration of myocardial interstitial inflammation cells in the AMI and ASD group (Figure 1E,G). No abnormalities were seen in the coronary arteries and myocardium in the sham group (Figure 1D,H). The postoperative electrocardiogram (ECG) showed that the ST segment was elevated in the ASD and AMI group compared with the sham group (Figure 1I–K). The results of the ECG showed that the ST amplitude was significantly increased and that the heart rate was significantly decreased in the AMI and ASD group (Figure 1M,N). The rat heart apex turned white after ligation (Figure 1L).

### 3.2. Differential Expression of miRNA in Rat Myocardium

Next, we measured the expression level of miRNA in rat cardiac tissue, and the TPM algorithm was used to normalize the expression level. Differentially expressed miRNAs (DEmiRNAs) were identified using edeR software (Appendix A, corrected *p* < 0.05). Volcano plots showed the differences in miRNA expression levels. Compared with the sham group, the ASD group had 217 miRNAs, with 147 miRNAs down-regulated and 70 miRNAs up-regulated (Figure 2A). The AS group had 224 miRNAs, with 121 miRNAs down-regulated and 103 miRNAs up-regulated (Figure 2B). The AMI group had 86 miRNAs, with 19 miRNAs down-regulated and 67 miRNAs up-regulated (Figure 2C). Similarly, the expression of miRNAs can be visually observed by using a heat map of the cluster analysis (Figure 2D–F).

### 3.3. Target Gene Analysis of Differential miRNA

Next, target gene prediction was performed using miRanda and TargetScan based on the gene sequence information. KEGG pathway enrichment analysis was performed on the differential target genes. The results showed that the ASD group was related to the biosynthesis of ubiquinone and other terpenoid-quinones, Ras, Oxytocin, thyroid hormone, the Rap1 signaling pathway, and others (Figure 3A). The AS group was mainly related to axonal guidance, mTOR, Hippo, the Ras and Rap1 signaling pathways, synaptic vesicle circulation, and steroid biosynthesis (Figure 3B). The AMI group was mainly associated with β-alanine metabolism, the synaptic vesicle cycle, the sphingolipid signaling pathway, sphingolipid metabolism, the Rap1 signaling pathway, and others (Figure 3C).

### 3.4. miRNA-mRNA Regulatory Network and Key miRNA or Genes

The miRNA-mRNA network showed that multiple target genes were regulated by one miRNA and that the same target gene was regulated by multiple miRNAs. We performed the mRNA sequencing of SCD rats in the previous stage. Next, we intersected the target genes of differential miRNAs with the differential genes obtained in the previous stage. Then, we searched for the opposite trend in miRNA and gene expression to construct the network (Figure 4A,B). In the miRNA-mRNA network, we found that miR542-Atg12 was involved in the RIG-I-like receptor signaling pathway, miR6328-Gstz1 was involved in tyrosine metabolism, and miR483-Dusp5 was involved in the MAPK signaling pathway. The 10 key genes were screened for the intersection of differential genes using Cytoscape, based on the cytohubba plugin (Figure 4C). To explore the specific miRNAs of each group, Venn diagrams were constructed to exclude common differential miRNAs. The results showed that there were 71 specific miRNAs in the ASD group, 76 specific miRNAs in the AS group, and 67 specific miRNAs in the AMI group (Figure 4D). LASSO regression was performed on the specific miRNAs screened in the ASD group and AMI group, and the key miRNAs in the ASD group were obtained: miR.106b, miR.195, miR.33, miR.652, miR.466b, and miR.6321. The key miRNAs in the AMI group were obtained: miR.205, miR.877, miR.325, and miR.344b (Figure 4E,F). Next, we plotted the ROC curves for the miRNAs selected by Lasso regression. The rate was 82% for the ASD group and 75% for the AMI group (Figure 4G,H).

## 4. Discussion

In the practice of forensic pathology, the diagnosis of SCD is difficult because, often, its morphology is not obvious or else it is associated with other diseases. Therefore, it is important to find new biomarkers for the diagnosis of SCD. Many studies have proven that miRNAs are involved in the regulation of multiple pathological processes of CAD. MiRNAs can regulate the gene expression of, for example, low-density lipoprotein receptor (LDLR), apolipoprotein B (APOB), and ATP-binding cassette transporter A1 (ABCA1). The overexpression of miR-148a in mice leads to an increase in circulating low-density lipoprotein cholesterol (LDL-C) and a decrease in plasma high-density lipoprotein cholesterol (HDL-C) levels [20,21]. The overexpression of miR-30c can reduce lipid synthesis and decrease the secretion of APOB, alleviating atherosclerotic lesions in ApoE-deficient mice [18]. Some miRNAs are involved in adipocyte differentiation, fatty acid synthesis, and the regulation of lipid metabolism-related genes, such as miR-33 and miR-122 [22]. It has been found that miR-223 plays a role in lipid metabolism, which increases in the liver of atherosclerotic mice by directly targeting and inhibiting scavenger receptor BI to regulate HDL-C uptake [23].

In this study, miR.106b, miR.195, miR.33, miR.652, miR.466b, and miR.6321 are related to ASD and can therefore be used as a diagnostic biomarker of ASD. These miRNAs have been studied to prove their relationship with CAD. MiR-106b-5p can be used to diagnose UA and STEMI, with a higher expression in STEMI compared to UA [24,25]. Li Jie et al. found that miR-195 may serve as a potential biomarker for coronary heart disease [26]. MicroRNA-195-3p could regulate P-ERK1/2 to maintain EC function under hypoxic conditions and thus serve as a new treatment strategy for CHDs [27]. Studies have also shown that the expression levels of plasma miR-33 are different in CAD patients and healthy controls [28]. Early studies have shown that miR-652 acts as a novel candidate biomarker for post-ACS prognosis [29]. However, miR-466b and miR-6321 have not been reported to be associated with ASD so far. This may be because the data came from animals rather than humans and because the sample size was small. The animal experiments we conducted might also affect the results of miRNA sequencing. In the future, we will use samples from corpses to detect and verify gene expression.

miR.205, miR.877, miR.325, and miR.344b were related to AMI and can thus be used as diagnostic biomarkers. It has been reported that miR-205-5p can increase the expression of Bax/Bcl-2 and Cleaved-caspase3 to promote cell apoptosis and inhibit proliferation and migration [30]. It has been demonstrated that miRNA-877-3p regulates HOXA-AS2 due to its proliferative and migratory abilities and promotes the apoptosis of vascular smooth muscle cells [31]. miR-325-3p can effectively ameliorate the symptoms of AMI by suppressing the expression of RIPK3 [32]. However, miR-344b was not related to AMI. Next, we will proceed to conduct the experimental verification.

Our research group had previously performed mRNA sequencing in SCD rats [19]. To find the potential mechanism of ASD, target genes of DEmiRNAs and differential genes were intersected. Next, an miRNA-mRNA regulatory network was constructed. It was found that miR542-Atg12 was involved in the RIG-I-like receptor signaling pathway, miR6328-Gstz1 was involved in tyrosine metabolism, and miR483-Dusp5 was involved in the MAPK signaling pathway. miR-542-3p exhibited the ability to protect against the pectoris of coronary heart disease by regulating GABARAP [33]. Another study showed that microRNA-520d-3p regulated Atg12 expression and affected cell viability, apoptosis, and autophagy, which play an important role in patients with CAD [34]. Li JB et al. showed that the miR-26a-5p/ATG12 axis regulates cardiomyocyte autophagy and apoptosis [35], while it has also been shown that the RIG-I-like receptor is involved in innate immunity, triggering immune responses [36]. Tyrosine metabolism is related to CAD, which is considered to be a potential biomarker of CAD [37,38]. However, miR6328-Gstz1 has not been reported to take part in ASD. This may be due to the variability of miRNA data sequencing results. In one study, MiR-483 was down-regulated in AMI and CAD patients compared with the controls, which suggests that it could be used as a biomarker of CAD [39]. Another study showed that miR-483 can suppress the expression of PCSK9 by directly binding to its 3 ′UTR, resulting in a reduction in LDL-C levels [40]. Dusp5 plays an important role in cardiac hypertrophy, which regulates the ERK1/2signaling pathway [41]. MiR-23a-3p could promote endothelial inflammation by targeting Dusp5 and maintaining ERK1/2 phosphorylation, which may transfer atherosclerosis to remote locations [42]. You HJ et al. found that the differentially expressed genes related to CAD are markedly abundant in the MAPK/cAMP/Ras signaling pathway [43], while another study suggested that the miR-363-3p-dependent NOX4 p38 MAPK axis serves as a promising target for CAD [44]. In the future, we will investigate the selected miRNA-mRNA regulatory network and signaling pathways further.

ASD and AMI were the most common causes of death in SCD. Therefore, this study simulated both causes of death. At the same time, we searched for their miRNA biomarkers and revealed the molecular mechanisms. These miRNAs presented a promising predictive biomarker for the forensic identification of SCD. The identified miRNA–target network and signaling pathways provide insights into the pathogenesis of SCD. However, the limitations of the study were as follows: (1) the sample size of the animal models was small, affecting the sequencing results; (2) no experiments were performed for further verification. In the future, the selected miRNAs and signaling pathways will be further verified by experiments.

## 5. Conclusions

MiR-106b, miR-195, miR-33, miR-652, miR-466b, and miR-6321 were identified as biomarkers in the ASD group, and miR-205, miR-877, miR-325, and miR-344b were identified as biomarkers in the AMI group. We found that miR542-Atg12 was involved in the RIG-I-like receptor signaling pathway, miR6328-Gstz1 was involved in tyrosine metabolism, and miR483-Dusp5 was involved in the MAPK signaling pathway. These miRNAs present a promising predictive biomarker for the forensic identification of SCD. The identified miRNA-target network and significantly activated pathways in this study provide insights into the pathogenesis of SCD.

## Figures and Tables

**Figure 1 cimb-47-00889-f001:**
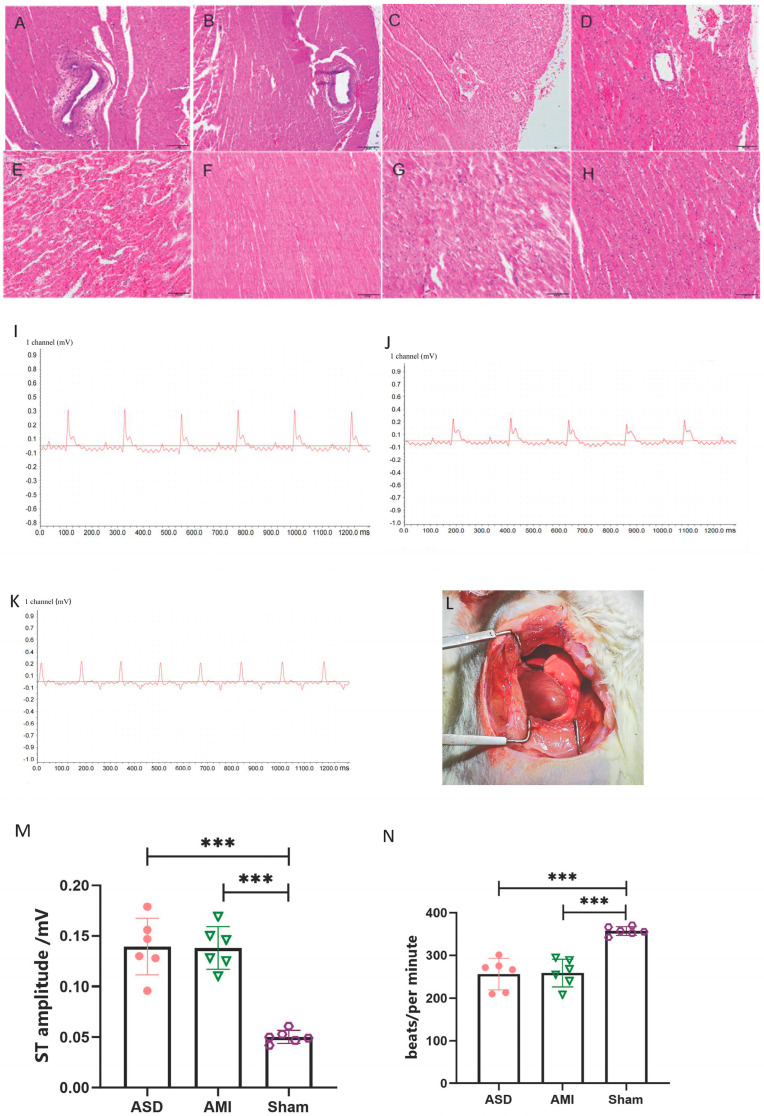
Pathological and electrocardiogram changes after the establishment of the rat model. (**A**–**D**). Pathological changes in coronary arteries in ASD, AS, AMI, and sham groups (200×). (**E**–**H**). Pathological changes in cardiomyocytes in ASD, AS, AMI, and sham groups (200×). (**I**). The electrocardiogram of the ASD group. (**J**). The electrocardiogram of the AMI group. (**K**). The electrocardiogram of the sham group. (**L**). Cardiac ischemia map after coronary artery ligation in rats. (**M**,**N**). Statistical difference chart of ST amplitude and bpm of ASD, AMI, and sham group. ASD: atherosclerotic death, AS: coronary atherosclerosis, AMI: acute myocardial ischemia. Data are pooled from two independent experiments. Data are shown as means ± SEMS; *** *p* < 0.001.

**Figure 2 cimb-47-00889-f002:**
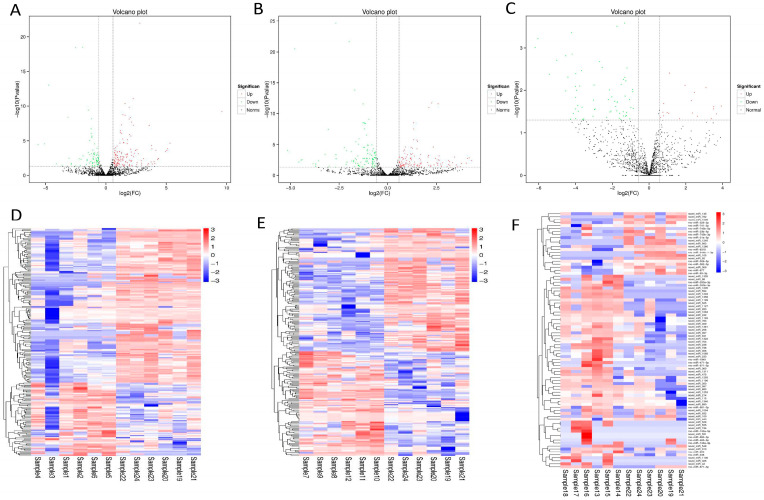
Differential expression of miRNA in rat myocardium. (**A**). Volcano plot of differentially expressed miRNAs in ASD group; (**B**). Volcano plot of differentially expressed miRNAs in AS group; (**C**). Volcano plot of differentially expressed miRNAs in AMI group. Up-regulated miRNA expression is shown in red, and down-regulated miRNA expression is shown in green. (**D**). Heat map of differentially expressed miRNAs in ASD group; (**E**). Heat map of differentially expressed miRNAs in AS group; (**F**). Heat map of differentially expressed miRNAs in AMI group. Up-regulated miRNA expression is shown in red and down-regulated miRNA expression is shown in blue. Sample 1–6: ASD group; Sample 7–12: AS group; Sample 13–18: AMI group; Sample 19–24: Sham group.

**Figure 3 cimb-47-00889-f003:**
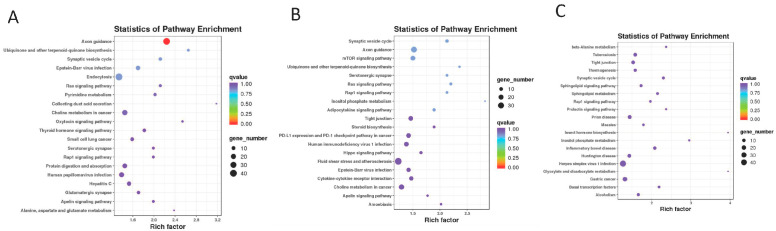
Target gene analysis of differential miRNA. (**A**). The KEGG analysis in the ASD group; (**B**). the KEGG analysis in the AS group; and (**C**). the KEGG analysis in the AMI group. The horizontal axis represents the enrichment factor, the vertical axis shows the pathways, and the size of the circles indicates the number of enriched genes, while the color of the circles represents the q-value.

**Figure 4 cimb-47-00889-f004:**
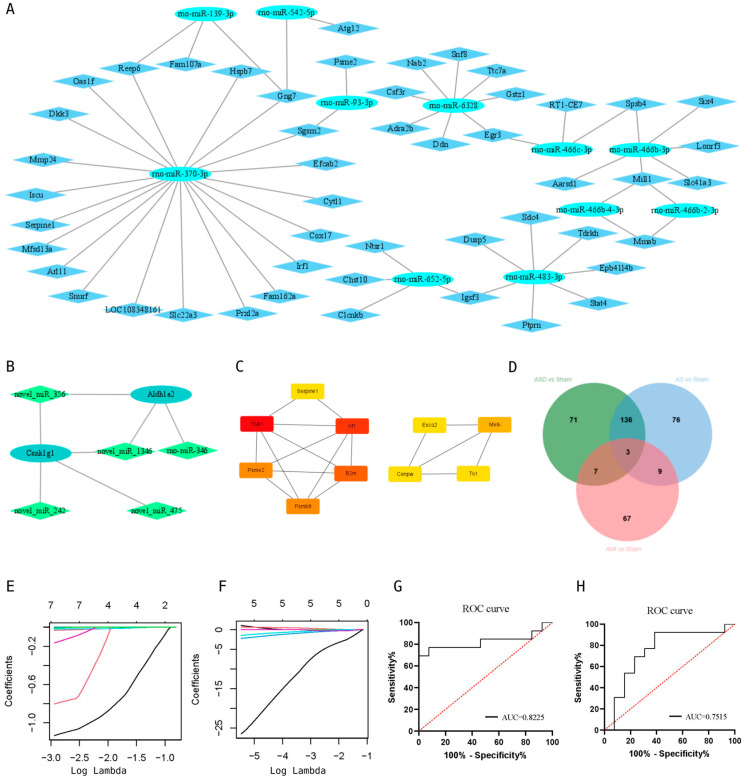
miRNA-mRNA regulatory network and identification of key miRNAs. (**A**). miRNA-mRNA regulatory network of the ASD group; (**B**). miRNA-mRNA regulatory network of the AMI group; (**C**). the top 10 key genes in the ASD group; (**D**). Venn diagram of differential miRNAs in the ASD group and AS and AMI group; (**E**). LASSO egression plot of the ASD group; (**F**). LASSO egression plot of the AMI group; (**G**). ROC curve graph of the ASD group; (**H**). ROC curve graph of the AMI group.

**Table 1 cimb-47-00889-t001:** Comparison of blood lipids and AI index in rats.

	AS/ASD Group	AMI/Sham Group
TC (mmol/L)	3.93 ± 0.56 *	1.96 ± 0.14
TG (mmol/L)	2.17 ± 0.24 *	1.52 ± 0.65
LDL (mmol/L)	1.07 ± 0.22 *	0.18 ± 0.07
HDL (mmol/L)	0.69 ± 0.12 *	0.5 ± 0.03
AI (mmol/L)	4.77 ± 0.47 *	2.92 ± 0.11

* *p* < 0.05, TC: total cholesterol, TG: Triglyceride, HDL: high-density lipoprotein; LDL: low-density lipoprotein, AI: atherosclerosis index, ASD: atherosclerotic death, AS: coronary atherosclerosis, AMI: acute myocardial ischemia.

## Data Availability

All data used for the analyses in this report are available from the corresponding author upon reasonable request.

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
