# Peer review of "Myocardium miRNA Analysis Reveals Potential Biomarkers of Sudden Coronary Death in Rats"

_cimb, 2025, doi:10.3390/cimb47110889_

Round 1
Reviewer 1 Report
Comments and Suggestions for Authors
Myocardium Transcript analysis reveals potential biomarkers of sudden coronary death in rats
Line 44: The study showed that the values of picolinic acid/Kynurenic acid and picolinic acid/3-hydroxyanthranillic acid maybe a biomarker for SCD in postmortem bodies [7] – please describe or give a hint on the study.
Line 74: Blood samples were taken from the orbit to measure the concentrations of four kinds of blood lipids (TC, TG, HDL, LDL) and observe the degree of atherosclerosis – How would you observe the degree of atherosclerosis through lipid profile, have you included any vascular studies?
Line 81: The library was constructed after the samples wres qualified. – do you mean were quantified and passed QC?
Line 82: Q-PCR – it is qPCR.
Line 88: to observe the degree of atherosclerosis and myocardial ischemia – what are the variable measured?
Line 103: paired or unpaired 2- tailed t test – which comparisons are done using paired T test?
Figure 1 : you name it as different groups- which group is each pic. representing please clearly label groups!
Where is I or L in the figure legend – where O-P in the figures – Please carefully correct figures and figure legend to reflect data, you are presenting.
Figure 2: No information could be taken out of it, Please include better resolution and names of dysregulated miRNAs!!!
Line 142: Next, target gene prediction was performed using miRanda and targetscan based on the gene sequence information. We found there were 3011 differential target genes in ASD group, 3388 differential target genes in AS group and 2407 differential target genes in AMI group. Where are these results – please include it at least as supplementary – score prediction and reference have to be included.
Figure 3 and 4: need better resolution.
Figure 4: need more explanation.
Thanks
Author Response
Thank you very much for taking the time to review this manuscript. We have revised and responded to your questions.

Reviewer 2 Report
Comments and Suggestions for Authors
The manuscript entitled ‘Myocardium Transcript analysis reveals potential biomarkers of sudden coronary death in rats’ written by Chunmei Zhao et al. presents interesting findings on identification of differentially expressed miRNAs with a diagnostic potential for detection of sudden cardiac death.
The authors performed comparative miRNA expression analysis between rat models of sudden cardiac death and control samples using RNA-seq experiments, and identified significantly differentially expressed miRNAs, their targeted genes and associated biological processes.
Although the research topic is clinically significant, as sudden cardiac death diagnosis is challenging in forensic pathology, the manuscript contains multiple methodological gaps, and the presentation of the results is of a unexpectedly low quality. Additionally, the editing and formatting layer of the manuscript has numerous errors; however, the English language is generally clear.
Below, I provide the most important comments and suggestions aimed at improving the scientific rigor and overall quality of the manuscript.
General concept comments
- The information provided in the Introduction section appears insufficient to establish a comprehensive background necessary for readers, particularly those who are not specialists in the field. I recommend the authors to briefly describe the pathophysiological and molecular mechanisms underlying the major clinical events leading to mortality in CAD and AMI. Clarifying the mechanistic links between these diseases and sudden cardiac death would substantially improve accessibility for a broader readership. In addition, a more detailed description of structural characteristics and molecular functions of miRNAs would provide essential molecular context and better emphasize the significance of the present study. More deep description of these aspects would enhance both the scientific justification and the rationale of the study.
- A significant limitation of the manuscript lies in the Materials and Methods section, which is presented in too concise manner and lacks essential details necessary to ensure the reproducibility of the study. Especially, there is no information about fundamental experimental procedures, including name of equipment used for RNA purity assessment, RNA purity criteria, names of kit used for libraries preparation for sequencing, name of used sequencing machine and kits, and software used for data processing. Furthermore, the information about quality control of RNA-seq data (MA plots, outlier identification) should be added. Experimental details of HE staining are also lacking. All these information are crucial for transparency and reproducibility of the study.
- The authors used P < 0.05 as a miRNA selection threshold. However, there is no information whether p values used for this threshold are simply raw p values or adjusted for multiple testing (e.g., using Benjamini-Hochberg or other method). P values adjustment is important for control type I error rate and reduce the likelihood of false positives.
- The study rigor would be improved by adding a Venn diagram showing common and specific miRNAs selected from three performed comparisons.
- As the study aimed to identify miRNAs indicative for studied diseases, I recommend the authors to extend their methodology by using ROC (Receiver Operating Characteristic) analysis, which enable to assess diagnostic value of selected miRNAs by calculating e.g., diagnostic sensitivity, specificity and area under ROC curves. Such results would significantly help the assessment of biomarker potential of selected miRNAs.
- I suggest the authors to enrich functional analysis by Gene Ontology analysis (Biological Process, Molecular Function, Cellular Compartment), which is a standard method for functional annotation of genes.
- In general, the figure legends are very brief and should be written in more detail to make the figures fully understand without reading the main text.
Specific comments
- I recommend revising the title to more accurately reflect the scope of the study. Specifically, it would be beneficial to indicate that the work involves the identification of miRNA biomarkers. The term “transcript” may not be the most precise descriptor in this context, and replacing it with “miRNA” could provide greater clarity and specificity.
- Please ensure that all abbreviations are explained when the first used in the text (e.g., lines 68, 69, 75, 77, …)
- In lines 109-111, the authors stated “The Atherogenic index (AI) was 4.77 in AS/ASD Group, indicating that the coronary atherosclerosis was successfully established”. However, it is not clear how atherogenic index was calculated. Additionally, the conclusion of the presence of atherosclerosis based only on elevated levels of lipids seems to be not justified. Although high levels of lipids increase risk of atherosclerosis, atherosclerotic plaques can only be detected when using imaging or arteries sections. Please revise this statement to avoid overstating.
- Table 1 – please add units of blood lipids levels and explain all used abbreviations.
- Legend to the Figure 1 is too brief, making the figure interpretation difficult and confusing. I suggest to add short description of pathological changes visible in panels A-L, together with clarification panel-group assignation. All abbreviations used in the figure and its legend should be explained. O and P panels mentioned in the figure legend are not present in the figure.
- Figure 2 – The specific comparisons illustrated in each panel are not clearly defined. Please provide unambiguous information, for example: “Panel A presents the ASD vs. sham comparison, Panel B depicts the AS vs. sham comparison,” etc. In addition, the heatmaps lack clear labeling of sample groups, making it difficult to determine which samples correspond to which experimental conditions. Please add annotation bar to the heatmap to assign each sample to proper group. Adding hierarchical clustering of samples (similarly as for miRNAs) would enrich the results interpretation. Finally, miRNA names are missing in panels D and E, and miRNA names font in panel F should be enlarged to improve readability.
- The font size for functional terms in Figure 3 is very small, which affects readability. The font should be enlarged to ensure the plot is clear and easy to interpret.
- For the study transparency, I strongly recommend to provide a table with differentially expressed miRNAs, together with their associated baseline expression, fold change values, p values, and adjusted p values. Due to the size of this table, this can be provided as supplementary table.
- The manuscript contains an unexpectedly high number of typographical, grammatical, and formatting errors (e.g., at lines 18, 69, 71, 81, and much more). A comprehensive and careful revision of the text is necessary to ensure clarity, accuracy, and adherence to scientific writing standards.
I believe that addressing the above points will enhance the clarity, precision, and presentation quality of the work.
Comments on the Quality of English LanguageThe manuscript would benefit from revision of the English language to achieve a more formal and academic style, consistent with standards of scientific writing.
Author Response

(The authors gave the same response as above.)

Reviewer 3 Report
Comments and Suggestions for Authors
The paper “Myocardium Transcript analysis reveals potential biomarkers of sudden coronary death in rats” presents the results of the analysis of differentially expressed microRNAs in the myocardium of three groups of rats with modeled signs atherosclerotic death, coronary atherosclerosis, acute myocardial ischemia in order to identify potential biomarkers of sudden coronary death.
The work consists of two parts - the creation of models simulating myocardial diseases and the search for biomarkers of sudden coronary death among miRNAs using bioinformatics methods.
The experimental design of the study is good, but the manuscript is carelessly formatted. There are quite a few points that require consideration and clarification.
The general impression is that there is little information in the Methods and Results sections, and the figures are also uninformative due to the lack of notations on them and captions under them.
Specific comments are as follows:
Abstract.
L13-16 – is this one sentence?
L19 – cytoscape or Cytoscape? Names are usually written with a capital letter
L17 – limma package is the only mention here. It is not in the Methods
L18 – TargetScan. But in Methods it is written with a lowercase letter
Methods
L75 – abbreviations about LIPIDS need to be deciphered. How were lipids determined? (reagents, specify company)
L75 – what means “observe degree of atherosclerosis”?
L79 – Trizol reagent - specifycompany cat. No.
L81-83 - need to describe Q-PCR, sequencing in more detail.
L97-100 - More details are also needed about bioinformatics tools lasso (Lasso), Cytoscape (version?), TargetScan (version?), miRanda
More details are also needed about Volcano, TPM algorithm, Heatmap, limma package, Venn diagram.
Results
It is necessary to describe Table 1 in the text.
L109 - it is not clear what AS/ASD group, AMI/Sham means – is it a sum? and/or?
L115 - "ST segment" but further on L125 – "ST amplitude". These are different concepts.
L128 - “we measured the expression level of miRNA of rat cardiac tissue”. It is necessary to present the results as Supplementary data.
L134 - Describe the results of Fig. 2 D-F, the groups are not indicated in the figures.
157-174 - It is necessary to make a section in more detail, describing ALL the figures. There is nothing in the text about Fig. 4B, for example. What does "we intersected the target genes of differential miRNAs with the differential genes" mean. Have you analyzed it? It is necessary to write the full name of all the mentioned genes
L170 - what is this group G?
Table and Figures
All figures must have a caption in addition to a brief explanatory title. Any special symbols or icons in the image must have a corresponding explanation in the caption.
In the captions to ALL figures and Table, decipher ALL abbreviations used in the figure, despite the presence of decipherments of similar abbreviations in the text
L116 -Table 1 - In the table title - clarify whether INDICATORS or indices are compared. It is necessary also to indicate the units of measurement. It is not clear what the numbers mean. It is necessary to make a list of abbreviations in the notes to the table.
Fig.1 A-H - Specify the magnification. Use arrows 1, 2, 3, etc. in the figures to indicate cells that are discussed in the text.
Fig. 1 I-K - Indicate the ST peaks in the figure.
Fig. 1L - no designation in caption.
Fig. 1 O-P (L125) is absent/
Fig. 2 - It is worth signing up for groups on Volcano plot and Heat map. It is better to use the term vs instead of between (L138, L139). In the caption, indicate what the columns and colors on Heat map mean.
Fig.4A - format is too small.
Fig.4C - it is not clear, here top key genes only for ASD group?
Fig.4D - it is not clear where which group is here. There should be designations in the figure.
Fig.4 E, F Check what is written on L179
Discussion
L219-221 - If sequencing was performed earlier, why is this not written in the Methods? This does not, however, exclude the need to present the information used for bioinformatics analysis as Supplementary data.
Author Response

(The authors gave the same response as above.)

Round 2
Reviewer 1 Report
Comments and Suggestions for Authors
No statistical test mentioned, the author mention Benjamini–Hochberg correction method but what is the test used in the first place, this is important!
Author Response
Thank you very much for taking the time to review this manuscript. For the question you raised, the following explanation is given. EdeR software was used for differential analysis of genes based on linear model. Negative binomial distribution model and different normalization methods were used to deal with gene differences between samples. According to the gene expression value, the t-test was used to identify the differentially expressed genes, and the p values were corrected for multiple testing using the Benjamini-Hochberg method.
Reviewer 2 Report
Comments and Suggestions for Authors
I appreciate the efforts of the authors to respond to my comments. Unfortunately, the responses from the authors are very limited and not in point-by-point structure, and with many comments and concerns not addressed.
Particularly, among general concept comments, comment no. 1 was not addressed at all. Comment no. 2 was addressed partially, only part of lacking information was added to the manuscript. Regarding comment no. 3, it is not clear what means asterisk before P - “∗P”. Furthermore, it seems that the authors used raw P value instead of adjusted P value, as from AMI vs. sham comparison, miRNA with raw P < 0.05 were selected, while no miRNA has adjusted P < 0.05.
The authors did not add ROC analysis results, as suggested in comment no. 5. Similarly, Gene Ontology functional analysis was not performed, as pointed in comment no. 6.
Regarding response to the specific comment no. 1, I do not agree with the authors to leave “transcript” expression in the title. This word suggest that the whole transcriptome was analyzed, while in fact only a minor part of cellular transcripts were investigated: short non-coding miRNA pool. Leaving “transcript” word could be confusing and do not precisely reflect the studied topic.
Although the authors provided the formula of Atherogenic Index calculation, they did not respond to my concerns regarding interpretation of the results. Atherogenic index in a biochemical parameter, which can be used to predict the risk of atherosclerosis, but it is not an evidence of presence of atherosclerotic lesions.
Regarding specific comment no. 5, O and P panels mentioned in the Figure 1 legend are still not present in this figure. Panel I is not referred in the Figure legend. It is still no clear which image is assigned to which group (AS, ASD, AMI, or sham).
Similarly, image-comparison assignation is not clear in Figure 2 (refer to specific comment no. 6). Simply listing comparisons in legend do not clarify this confusion. Sample annotation bar, hierarchical clustering and miRNA names were not added to heatmaps, as suggested.
Font size in Figure 3 was not enlarged, as mentioned in specific comment no. 7.
Due to all these severe concerns, I recommend the Editor to reject the manuscript from publication. However, I leave the final decision to the Editor, and I express my willing to cooperate further if the manuscript will be subjected to next round of revision. As the study retains scientific potential, I still hope that when all abovementioned issues will be addressed in the manuscript or the authors explain the reasons of rejections of my suggestions, the manuscript would be published.
Author Response
We are grateful for the invaluable feedback provided. We sincerely appreciate you the time and effort invested in evaluating our manuscript.

Reviewer 3 Report
Comments and Suggestions for Authors
The article may be accept in present form
Author Response

(The authors gave the same response as above.)

Round 3
Reviewer 2 Report
Comments and Suggestions for Authors
I appreciate the efforts of the authors to respond to my comments. The introduced changes partially adhered to my comments and improved the manuscript. I strongly recommend the authors to use point-by-point structure of their responses with indicating lines where the text was changed. It makes much easier to follow the revisions by reviewers.
Below I summarized the author responses to my comments (numbered as in the first review report) and highlighted some points that need further attention.
General concept comments
- Addressed.
- Some important information is still lacking. Please provide name of device used for RNA integrity analysis and RIN assessment. Please add name of kit used for libraries preparation for sequencing, name of model of used sequencer (like NextSeq 550 System, MiniSeq System, etc.). Please add information if the identification of outlier samples in miRNA expression data was performed or not.
- Thank you for the explanation regarding p value correction. For clarity, I suggest to re-write the text in lines 139-143 “In experiments comparing 2, statistical differences between groups were determined using a Student t test. Differences were considered significant at ∗P < 0.05. The Benjamini–Hochberg correction method was used to adjust the original hypothesis test results. Finally, the False Discovery Rate (FDR) was used to screen differentially expressed miRNAs.” to “In experiments comparing 2 groups, statistical differences between groups were determined using a Student t test. The Benjamini–Hochberg correction method was used to adjust the original hypothesis test results obtained from miRNA comparative analyses. miRNAs with adjusted P values (False Discovery Rates, FDR) < 0.05 were considered statistically significant.” or similar. Additionally, in each place in the manuscript where p value is mentioned (text, tables and figures), please clearly indicate whether it refers to adjusted p value. Additionally, on the volcano plots, the raw (not adjusted) p values are still presented. Please replaced volcano plots with those presenting adjusted p values.
- Addressed.
- Explained.
- I suggest the authors to shortly mention that no statistical significant functional terms were obtained from Gene Ontology analysis. The most relevant functional terms with the lowest p values can be shortly listed in the main text. For full transparency, the obtained results can be provided as supplementary tables.
- Addressed.
Specific comments
- Addressed.
- Addressed.
- Explained.
- Addressed.
- Addressed.
- Addressed and explained.
- Explained.
- Addressed.
- The manuscript still contains some grammatical and formatting errors (e.g., at lines 75-83, 100, 139). Please carefully check the language layer of the manuscript once again.
Author Response
Thank you very much for taking the time to review this manuscript. Thank you for your comments to make this article better read. We acknowledge the need for major revisions and are committed to addressing the highlighted issues comprehensively. We have revised and responded to your questions. At the same time, we have refined the article for better reading. All changes made have been noted.

Round 4
Reviewer 2 Report
Comments and Suggestions for Authors
Dear Authors,
Your response and subsequent revision do not correspond to the points I raised in my previous report. This may have occurred due to an inadvertent oversight. I kindly request you to provide a direct response to points 2, 3, and 6 from the general concept comments, as well as point 9 from the specific comments in my last report.
Author Response
Thank you very much for taking the time to review this manuscript. Thank you for your comments to make this article better read.
We sincerely appreciate you the time and effort invested in evaluating our manuscript.Thank you for your comments to make this article better read. All changes made have been noted. We look forward to any additional feedback or suggestions. We have worked hard to incorporate your feedback and hope that these revisions persuade you to accept our submission. The following is the response to your question.
- A significant limitation of the manuscript lies in the Materials and Methods section, which is presented in too concise manner and lacks essential details necessary to ensure the reproducibility of the study. Especially, there is no information about fundamental experimental procedures, including name of equipment used for RNA purity assessment, RNA purity criteria, names of kit used for libraries preparation for sequencing, name of used sequencing machine and kits, and software used for data processing. Furthermore, the information about quality control of RNA-seq data (MA plots, outlier identification) should be added. Experimental details of HE staining are also lacking. All these information are crucial for transparency and reproducibility of the study.
Answer:Thank you for your reply. We have provided a detailed description of the method. Although we did not provide detailed descriptions of the specific animal modeling methods in the article, We have cited the references for readers. We have provided detailed descriptions of RNA purity assessment, RNA purity criteria, names of kit used for libraries preparation for sequencing, name of used sequencing machine and kits, and software used for data processing. Furthermore,we also provided the information about quality control of RNA-seq data and HE staining.
- The authors used P < 0.05 as a miRNA selection threshold. However, there is no information whether p values used for this threshold are simply raw p values or adjusted for multiple testing (e.g., using Benjamini-Hochberg or other method). P values adjustment is important for control type I error rate and reduce the likelihood of false positives.
Answer:Thank you for your reply. We have already mentioned the p values in Sections 2.8 and 2.9. The Benjamini–Hochberg correction method was used to adjust the p values. The differential miRNAs was identified between the groups using the following screening criteria: |log2(FC)|≥0.58 and corrected P value ≤0.05. The p-values in the attached table are the corrected p-values.
- I suggest the authors to enrich functional analysis by Gene Ontology analysis (Biological Process, Molecular Function, Cellular Compartment), which is a standard method for functional annotation of genes.
Answer:Thank you for your reply. Gene Ontology analysis and KEGG pathway are a standard method for functional annotation of genes. The GO Ontology analysis results in ASD and AMI group were similar. The predicted differential target genes are involved in various biological processes such as cellular processes, single biological processes, biological regulation, metabolic processes, etc.; cellular components include cells, cell parts, organelles, membranes, etc.; molecular functions include binding, catalytic activity, transport activity, molecular function regulation, etc. KEGG pathway enrichment can better demonstrate the differences in signaling pathways present in the ASD and AMI group. Therefore, we only chose to present the results using KEGG pathway enrichment analysis.
- The manuscript contains an unexpectedly high number of typographical, grammatical, and formatting errors (e.g., at lines 18, 69, 71, 81, and much more). A comprehensive and careful revision of the text is necessary to ensure clarity, accuracy, and adherence to scientific writing standards.
Answer: Thank you for your understanding. We have already checked the manuscript. The manuscipt has undergone English language marked in yellow editing by MDPI for better reading(english-100236). The typographical, grammatical, and formatting errors were corrected. It was edited to a level suitable for reporting research in a scholarly journal.